# Effects of Sulfur Application on the Quality of Fresh Waxy Maize

**DOI:** 10.3390/plants13192677

**Published:** 2024-09-24

**Authors:** Chenyang Jiang, Yuwen Liang, Yuru Wang, Genji You, Jian Guo, Dalei Lu, Guanghao Li

**Affiliations:** 1Jiangsu Key Laboratory of Crop Genetics and Physiology/Jiangsu Key Laboratory of Crop Cultivation and Physiology, Yangzhou University, Yangzhou 225009, China; 13646104102@163.com (C.J.); lyw17866710733@163.com (Y.L.); 17851971675@163.com (Y.W.); a2728152904@163.com (G.Y.); guojian90816@126.com (J.G.); dllu@yzu.edu.cn (D.L.); 2Jiangsu Co-Innovation Center for Modern Production Technology of Grain Crops, Yangzhou University, Yangzhou 225009, China

**Keywords:** pasting property, sulfur, starch, thermal property, waxy maize flour

## Abstract

Balanced fertilizer application is crucial for achieving high-yield, high-quality, and efficient maize cultivation. Sulfur (S), considered a secondary nutrient, ranks as the fourth most essential plant nutrient after nitrogen (N), phosphorus (P), and potassium (K). S deficiency could significantly influence maize growth and development. Field experiments were conducted in Jiangsu, Yangzhou, China, from April 1 to July 20 in 2023. Jingkenuo2000 (JKN2000) and Suyunuo5 (SYN5) were used as experiment materials, and four treatments were set: no fertilizer application (F_0_), S fertilizer application (F_1_), conventional fertilization method (F_2_), and conventional fertilization method with additional S application (F_3_). The objective was to investigate the impact of S application on grain weight and the quality of fresh waxy maize flour and starch. The results indicated that all fertilization treatments significantly increased grain weight and the starch and protein contents in grains compared to no fertilization. Among these, F_3_ exhibited the most significant increases. Specifically, in JKN2000, the grain weight, starch content (SC), and protein content (PC) increased by 27.7%, 4.8%, and 14.8%, respectively, while in SYN5, these parameters increased by 26.3%, 6.2%, and 7.4%, respectively, followed by F_2_ and F_1_. Compared to F_0_, F_3_ increased starch and protein contents by 4.8% and 14.8% in JKN2000, and by 6.2% and 7.4% in SYN5. Compared to F_0_, F_2_ and F_3_ significantly increased the iodine binding capacity (IBC) of SYN5, with F_3_ being more effective than F_2_, while they had no significant effect on the IBC of JKN2000. The peak viscosity (PV) and breakdown viscosity (BD) of waxy maize flour and starch for both varieties showed a consistent response (increasing trend) to S application, and F_3_ had the largest increase. Regarding the thermal properties of waxy maize flour, F_3_ significantly enhanced the retrogradation enthalpy (Δ*H*_gel_) of both varieties compared to F_0_, while achieving the lowest retrogradation percentage (%*R*). In starch, the highest Δ*H*_gel_ and the lowest %*R* were observed under the F_2_ treatment. In summary, under the conditions of this experiment, adding S fertilizer to conventional fertilization not only increased the grain weight of waxy maize but also effectively optimized the pasting and thermal properties of waxy maize flour and starch.

## 1. Introduction

Waxy maize, also known as sticky maize, is valued for its rich nutritional content and palatability, and it covers over 800,000 ha in China [1]. The starch of waxy maize is composed almost entirely of amylopectin [2], resulting in high viscosity, low retrogradation, and easy digestibility. These properties made waxy maize starch widely used in both food and certain non-food industries [3]. As living standards improve, the market prospects for waxy maize are expanding both domestically and internationally. The physicochemical properties of its grains and starch are crucial in determining its edibility quality [4]. Sulfur, an essential mineral nutrient for plant growth, plays a critical role in physiological and biochemical functions for multiple crops [5]. Due to reduced atmospheric sulfur inputs, stricter environmental regulations, and changes in fertilization practices, sulfur deficiency is becoming a growing concern, and has varying impacts on different crops, including maize [4,6,7]. In recent years, unbalanced S fertilization has been observed in major maize-producing countries, including the United States, China, and Argentina, leading to yield losses and environmental issues [8,9].

Reasonable fertilization is crucial for enhancing crop yield and quality while safeguarding the ecological environment. Conversely, excessive, insufficient, or imbalanced fertilizer use can result in significant negative impacts. In recent decades, there has been a pronounced reliance on nitrogen (N) fertilizers to boost grain yields and protein content (PC), often overshadowing the potential benefits of sulfur (S) fertilizers in improving the quality of maize. Both N and S are indispensable nutrients necessary for plant growth and crop production [10,11]. Optimizing the balance of N and S enhanced crop productivity. N plays a pivotal role in crop essential physiological processes, including the formation of enzymes, hormones, and amino acids [12]. S is essential for the synthesis of proteins, chlorophyll, enzymes, and vitamins, impacting overall metabolic and photosynthetic processes [13]. Waxy maize flour, produced by grinding waxy maize grains into a fine powder, contains starch, protein, fiber, and other components, offering high nutritional value. Previous studies demonstrated that optimizing the application of NPK fertilizers made for higher pasting temperature (*P*_temp_), moderate gelatinization enthalpy (∆*H*_gel_), and retrogradation percentage (%*R*) in waxy maize flour [14]. Moreover, the combined application of N and K fertilizers increased the PC of the grains and enhanced the ∆*H*_gel_ of waxy maize flour [15]. Proper S fertilization can address the rapid growth needs of crops under high nitrogen conditions and enhance their yield [16]. The combined application of N and S fertilizers enhanced N uptake in maize, increasing yield, PC, and starch content (SC) in the grain s [17]. Compared to normal maize starch, waxy maize starch gelatinized more readily and produced a starch paste with superior transparency, stability, extensibility, and reduced retrogradation [18]. The application of S fertilizer significantly increased grain PC and altered the disulfide bond content in flour, thereby affecting the viscosity of the flour. However, the overapplication of S interfered with the uptake of other essential nutrients, which may impede protein accumulation [19,20]. Although N fertilizers are essential for increasing crop yield [21], optimal N levels improve the starch’s pasting and thermal properties [22,23]. Conversely, excessive N application can significantly reduce the PV (peak viscosity) and BD (breakdown viscosity) of waxy maize starch and increase *P*_temp_ [24]. And, a balanced application of NS fertilizers is more effective in enhancing both crop yield and quality. This is due to the interaction effects between N and S, as appropriate application of both fertilizers significantly enhanced the growth and development of maize [25]. Research indicated that crops exhibited greater responsiveness to S fertilizer when N supply was ample [26]. N fertilizer facilitated the accumulation of waxy maize starch, making the starch structure more stable (with higher Δ*H*_gel_), increasing starch PV, FV (final viscosity), SB (setback viscosity), and BD, and reducing starch %*R* [27,28]. Similarly, buckwheat exhibits a notable response to N fertilization. Research has demonstrated that optimal N application not only enhances buckwheat grain weight and PC but also improves starch quality [29,30]. On the basis of reasonable fertilization (N, P, K), additional S fertilizer effectively increased maize grain weight and grain PC, thereby improving quality [17,31,32]. Applying N and S fertilizers appropriately could enhance rapeseed yield, preserve its PC, and minimize fertilizer waste [33].

N fertilizers are commonly prioritized over sulfur in glutinous maize production due to their proven ability to boost yields and deliver significant short-term economic benefits. So, in maize production, there was a widespread phenomenon of emphasizing N while neglecting S fertilization, and the fertilization practice often lacked precision. Although the growth, yield, and quality of maize depend on N inputs, balanced fertilizer application is more beneficial for enhancing crop yield and quality [34]. It was a decisive factor in achieving S fertilizer response to apply S and N fertilizers to ensure a balanced nutrient supply. The response to S fertilizer increased with the amount of N fertilizer applied [35]. Previous research primarily focused on the effects of S fertilizer on crop yield and fertilizer use efficiency [36], but few studies have investigated the effects of S on the quality of fresh waxy maize flour and starch. Therefore, the aims of this study were (1) to investigate the effects of S application on the grain weight and quality-related indices of fresh waxy maize; (2) to study the response of fresh waxy maize flour and starch quality to S fertilizer; and (3) to develop guidelines for optimizing fertilization practices and improving the quality of fresh waxy maize in southern China.

## 2. Results

### 2.1. Grain Weight

The application of S fertilizer significantly influenced 100-grain weight and grain moisture content (Figure 1). The 100-grain weight of JKN2000 was higher than that of SYN5. Fertilization treatments resulted in significant increases in the 100-grain weight of both varieties compared to F_0_, while the moisture content did not exhibit statistically significant differences. The 100-grain weight of JKN2000 increased by 11.3% under F_1_, 24.0% under F_2_, and 27.4% under F_3_. For SYN5, the increases were 12.7% under F_1_, 26.3% under F_2_, and 27.7% under F_3_. The largest increase was observed with the F_3_ treatment.

### 2.2. Grain Component Content

In the single-factor analysis, variety (V) significantly influenced the starch and soluble sugar contents, while treatment (T) significantly impacted the starch, soluble sugar, and protein contents. The interaction (V × T) significantly affected the soluble sugar content but did not have a significant effect on starch and protein contents (Table 1). The SC, SSC (soluble sugar content), and PC of JKN2000 were 14.6%, 1.0%, and 1.4% higher, respectively, than those of SYN5, but the differences in protein were not statistically significant. Relative to the F_0_ treatment, the F_2_ and F_3_ treatments significantly increased the starch content and decreased the soluble sugar content. Both varieties showed the highest starch and protein contents and the lowest soluble sugar content under the F_3_ treatment.

### 2.3. Pasting Properties of Waxy Maize Flour and Starch

S fertilizer application significantly influenced the pasting properties of fresh waxy maize flour and starch (Figure 2). The PV, BD, and SB of waxy maize flour were greatly affected by the interaction between variety and treatment (V × T), as were the pasting property values of starch (Table 2). The PV and BD of the starch were 75.6% and 950.3% higher, respectively, compared to those of waxy maize flour. The PV, TV, BD, FV, and SB of waxy maize flour in JKN2000 were higher than those of SYN5, while the *P*_temp_ was lower than that of SYN5. the PV, BD, FV, and SB of starch in JKN2000 were higher than those of SYN5, with a lower *P*_temp_, and no significant difference in TV (trough viscosity) between the two varieties. The PV and BD of both waxy maize flour and starch showed an increasing trend under fertilization treatments, as follows, F_0_ < F_1_ < F_2_ < F_3_, and the increase in the PV and BD in waxy maize flour under F_3_ exceeded that observed in starch. However, the BD of JKN2000 starch showed no significant difference between F_1_ and F_2_ treatments. The PV and BD of SYN5 waxy maize flour showed no significant difference between F_1_ and F_2_ treatments, and the PV and BD of starch showed no significant difference between F_0_ and F_1_ treatments.

### 2.4. Thermal Properties of Waxy Maize Flour and Starch

The application of S fertilizer significantly affected the thermal properties of fresh waxy maize flour and starch (Table 3). The interaction between V and T had a significant impact on both the *T*_p_ and *T*_c_ of waxy maize flour and the gelatinization temperature (*T*_o_, *T*_p_, and *T*_c_) of starch. The Δ*H*_gel_ and ΔH_ret_ values of waxy maize flour were lower than those of starch, while its gelatinization temperatures (*T*_o_, *T*_p_, and *T*_c_) and *%R* were higher. The ∆*H*_gel_ and ∆*H*_ret_ of JKN2000 were lower than those of SYN5, with *%R* showing JKN2000 < SYN5 in waxy maize flour, whereas the opposite was found in starch. Fertilization treatments significantly increased Δ*H*_gel_ (reaching its maximum in F_3_) and decreased Δ*H*_ret_ and *%R* (reaching its minimum in F_3_) in waxy maize flour; similar effects were found in F_2_ for starch.

### 2.5. Maximum Absorption Wavelength and Iodine Binding Capacity of Waxy Maize Starch

The application of S fertilizer significantly affected the iodine binding capacity (IBC) and maximum absorption wavelength (MAW) of fresh waxy maize starch (Figure 3). The IBC of JKN2000 was lower than that of SYN5, but its MAW was higher, with no significant difference observed between them. Fertilizer treatments did not significantly affect the IBC of JKN2000 or the MAW of SYN5. Compared to F_0_, the IBC of SYN5 significantly increased in F_2_ and F_3_, with a greater increase in F_3_. The MAW of JKN2000 significantly decreased across F_1_, F_2_, and F_3_ treatments, with no significant differences observed among these treatments.

### 2.6. Correlation Analysis

#### 2.6.1. Correlation Analysis of Grain Quality and Component Contents of Waxy Maize

The results of correlation analysis (Figure 4) on waxy maize flour quality showed that SC was positively correlated with PC, PV, TV, BD, FV, SB, and Δ*H*_gel_, but negatively correlated with SSC and *%R*. SSC was positively correlated with *%R*, but negatively correlated with PC and Δ*H*_gel_. PC was positively correlated with PV, TV, BD, FV, and Δ*H*_gel_, but negatively correlated with *%R*. PC, PV, TV, BD, and FV were positively correlated with each other pairwise. *T*_o_, *T*_p_, and T_c_ showed positive correlations pairwise.

#### 2.6.2. Correlation Analysis of Starch Quality of Waxy Maize

The results of correlation analysis (Figure 5) on starch quality showed that IBC was positively correlated with PV, TV, FV, SB, and ΔH_gel_, but negatively correlated with *%R*; the MAW was positively correlated with *P*_temp_, *T*_o_, and *T*_p_. PV, TV, BD FV, and SB were positively correlated with each other pairwise. *T*_o_, *T*_p_, and *P*_temp_ were positively correlated with each other pairwise. 

## 3. Discussion

One of the key cultivation factors for increasing crop yield was fertilization [37]. Implementing scientifically based and rational fertilization practices is essential for maximizing maize yield, improving quality, and increasing overall efficiency. Zhao et al. [38] found that S fertilizer application significantly increased grain weight. This increase could be attributed to several factors. The application of S fertilizer increased glutathione levels in maize leaves, which reduced hydrogen peroxide accumulation [39]. This reduction helped maintain redox balance during photosynthesis [40], leading to an improved photosynthetic rate [41] and enhanced production of photosynthetic assimilates. The increase in photosynthetic products enhanced dry matter accumulation, which subsequently resulted in a substantial increase in maize grain weight. Similar results were observed in this experiment. The synergistic effect of N and S fertilizers significantly enhanced maize growth and development [42]. Compared to no fertilizer application, all fertilization treatments markedly increased the grain weight of fresh waxy maize (F_1_ increased by 0.4%, F_2_ increased by 37.7%, and F_3_ increased by 54.2%), as well as the PC and SC in the grains. Notably, the conventional fertilization method with additional S application (F_3_) outperformed both the S fertilizer application (F_1_) and the conventional fertilization method (F_2_) in increasing these parameters.

Maize grains are primarily composed of starch, proteins, and soluble sugars, which influence the physicochemical properties of waxy maize flour [2]. The quality of grains was profoundly impacted by environmental conditions and fertilization practices [43]. Balanced fertilization increased the SC and PC in grains [44]. Appropriate S fertilizer application enhanced the PC in maize grains [20], as confirmed by this study. S fertilizer application significantly increased the PC and SC in grains. The increase was likely due to S fertilizers promoting the cross-linking of cysteine and other amino acids, forming disulfide bonds that maintain protein stability and promote protein accumulation [45]. Protein synthesis requires the combined action of N and S [46]. Previous studies have shown that their combined application increased grain PC [47]. This study further corroborates these findings. Both N and S play essential roles in protein synthesis, with N serving as a primary building block of amino acids and S supporting the synthesis of key sulfur-containing amino acids like cysteine and methionine [48]. The synergistic application of these two elements not only optimizes the amino acid composition but also enhances protein quality. Furthermore, S facilitates N uptake and utilization in crops, improving N use efficiency and subsequently increasing PC [49], so the conventional fertilization method with additional S application (F_3_) led to a more substantial increase in SC and PC than the S fertilizer application (F_1_) and the conventional fertilization method (F_2_).

The pasting and thermal properties served as crucial indicators for assessing the quality of fresh waxy maize. Adding S fertilizer altered the content of disulfide bonds in wheat flour, thereby affecting its viscosity [20], and similar effects might be expected in maize. Previous studies found that changes in the composition of waxy maize grains affected the PV of waxy maize flour [50]. The results of this study indicated that compared to F_0_, fertilization significantly increased the SC and PV in grains of both varieties, with the highest values observed under F_3_. The increased SC enhanced the swelling tendency of starch granules, allowing them to fully absorb water and swell, thereby increasing the pasting viscosity [51]. In food processing, higher viscosity helps in forming a uniform food structure. This leads to improved appearance and stability of the final product, as well as reduced layering or separation during storage and transportation [52]. Correlation analysis of waxy maize flour quality showed that SC was significantly positively correlated with Δ*H*_gel_, and significantly negatively correlated with %*R*. This indicated that a high SC was conducive to a high Δ*H*_gel_ (better thermal stability). Under the experimental conditions, fertilization treatments led to a significant increase in starch content in both maize varieties. Waxy maize starch, consisting of nearly 100% amylopectin, requires more energy to disrupt and gelatinize its highly ordered structure. This results in greater thermal stability and improved anti-retrogradation properties [53]. Consequently, sulfur application in this study enhanced the starch content of waxy maize, leading to an increase in Δ*H*_gel_. Previous research showed that waxy maize flour with high Δ*H*_gel_ often formed higher pasting viscosity after gelatinization [54]. This study had similar results, indicating that a high Δ*H*_gel_ was associated with stronger hydrogen bonds in starch granules, necessitating more energy to break these bonds during gelatinization and resulting in higher pasting viscosity [55]. Moreover, prior studies demonstrated that the Δ*H*_gel_ value of starch was higher than that of waxy maize flour [56], a finding corroborated by this research. The higher Δ*H*_gel_ of starch was typically related to its greater crystallinity and more complex molecular structure. In contrast, waxy maize flour has lower crystallinity and a looser structure, while starch, with its higher crystallinity, requires more thermal energy to overcome its crystalline structure and complete gelatinization. Consequently, the Δ*H*_gel_ of starch was higher than that of waxy maize flour [57].

N fertilization changed the pasting and thermal properties of starch [27]. Previous research showed that the incorporation of S fertilizer alongside N improved the pasting viscosity (PV, TV, FV) and BD of wheat starch [58]. These findings are similar to the results of our study, which showed that compared to F_1_ and F_2_, F_3_ notably elevated the PV, BD, and *P*_temp_ of waxy maize starch. Correlation analysis of starch quality showed that IBC was significantly positively correlated with pasting viscosities (PV, TV, FV) and Δ*H*_gel_, and significantly negatively correlated with %*R*. This suggests that a higher IBC enhances starch’s swelling and shear resistance, leading to increased pasting viscosity and improved anti-retrogradation properties [28]. Under the conditions of this experiment, SYN5 exhibited a more significant response to fertilization regarding IBC, with the highest value recorded under the F_3_ treatment. The increased IBC may result from fertilization, which reduced the activity of starch branching enzymes. This reduction led to decreased branching frequency in amylopectin, resulting in a higher proportion of long chains. These longer chains could form more stable complexes with iodine, thereby enhancing the IBC [59]; amylopectin with long chains has higher pasting viscosity after gelatinization, higher Δ*H*_gel_, better thermal stability, and stronger anti-retrogradation ability [60].

## 4. Materials and Methods

### 4.1. Experimental Design

Field experiments were conducted in 2023 at the Yangzhou University experimental farm in Yangzhou, Jiangsu Province, China (32.40° N, 119.43° E). Jingkenuo2000 (JKN2000) and Suyunuo5 (SYN5) were used as experimental materials. In southern China, SYN5 was utilized as the benchmark variety for fresh waxy maize regional trials, whereas JKN2000 stand out as the predominant waxy maize cultivar across the nation. The test soil under examination predominantly exhibited a silt loam texture, and the PH was 6.2. The nutrient composition of the topsoil (0–20 cm) prior to sowing was as follows: average contents of organic matter 17.1 g kg^−1^, total N 1.0 g kg^−1^, alkali hydrolysable N 103.0 mg kg^−1^, available phosphorus 125.6 mg kg^−1^, available potassium 136.6 mg kg^−1^, and total sulfur 292.4 mg kg^−1^.

The field experiment employed compound fertilizer (N−P_2_O_5_−K_2_O = 27−9−9, provided by Jiangsu Zhongdong Fertilizer Co., Ltd., Changzhou, China) and sulfur fertilizer (sulfur content 95%, bought from Tiger-Sul, Inc., Shelton, CT, USA) as sources of N, P, K, and S fertilizers. This experiment comprised four treatments: no fertilizer application (F_0_), S fertilizer application (F_1_, 75 kg S ha^−1^), conventional fertilization method (F_2_, N/P_2_O_5_/K_2_O = 225/75/75 kg ha^−1^), and conventional fertilization method with additional S application (F_3_, N/P_2_O_5_/K_2_O/S = 225/75/75/75 kg ha^−1^), with 3 replicates for each treatment. At sowing, all fertilizers were uniformly applied as basal dressing, and other agronomic practices (water management, pest and disease control) followed high-yield management protocols.

Each plot, covering an area of 30 m^2^ (5 m × 6 m), was cultivated with a planting density of 60,000 plants ha^−1^, utilizing a double-row (0.8 m and 0.4 m) planting method. Maize was sown on April 1 and harvested on July 20, with a growth period of 111 days. The meteorological conditions of the field experiment are shown in Figure 6. The effective accumulated temperature, total rainfall, and total sunlight hours in the maize (*Zea mays* L.) growing seasons were 1401.9 °C, 698.0 mm, and 530.9 h. 

### 4.2. Grain Weight

During the milk stage, we selected three sample points for each treatment. At each of these sample points, we employed a continuous sampling method to collect 10 representative ears. Subsequently, we determined the 100-grain weight and grain moisture content for each of these samples.

### 4.3. Grain Component Content

The soluble sugar and starch contents in the grains were assessed through the anthrone–sulfuric acid method [61], while the nitrogen content was measured using the Kjeldahl method. Protein content was calculated as nitrogen content × 6.25.

### 4.4. Preparation of Waxy Maize Flour and Starch Samples

Each treatment selected three uniformly developed maize ears, from which grains were carefully extracted from the middle and thoroughly mixed. Subsequently, 200 g of grains were randomly sampled for the preparation of starch and waxy maize flour samples. The grains (100 g) were steeped in 500 mL of 1 g L^−1^ NaHSO_3_ solution for 48 h at room temperature. Following this, starch separation was carried out following established protocols from previous studies [18]. The grains (100 g) were subjected to drying in an oven (TENGREN DZ47–63) set at 60 °C until reaching a constant weight. After drying, the grains were finely ground using a grinder (RS–FS1406) and passed through a 100-mesh sieve. The resulting grain flour was utilized for analyzing grain flour component content, gelatinization, and thermodynamic properties.

### 4.5. Pasting Property and Thermal Property of Waxy Maize Flour and Starch

Using waxy maize flour and starch as experimental materials, the gelatinization properties were determined following established methods [62]. Specifically, 2.8 g of maize flour was combined with 25.2 g of ultrapure water to create a 10% concentration maize slurry. Similarly, 1.96 g of starch was mixed with 26.04 g of ultrapure water to prepare a 7% concentration starch slurry. Subsequently, the gelatinization properties of the starch (total weight 28 g; 7% *w*/*w*, dry basis) and maize flour were analyzed using a Rapid Visco Analyzer (RVA, Model 3D, Newport Science, Warriewood NSW, Australia). For the determination of thermodynamic properties, 5 mg of dried sample was placed into a small crucible (25/40 µL, D = 5 mm), and an equivalent amount of ultrapure water was added to achieve a 66.7% water–starch suspension. Each suspension was sealed under pressure, and three replicates were prepared for each treatment. Following a 24-h storage period at 4 °C, thermodynamic properties were measured using a Differential Scanning Calorimeter (DSC, Model 200 F3 Maia, NETZSCH, Bavaria, Germany). After measurement, the samples were refrigerated at 4 °C, and retrogradation percentages were assessed after 7 days.

### 4.6. Maximum Absorption Wavelength and Iodine Binding Capacity of Waxy Maize Starch

The UV–Vis spectrophotometer model utilized to determine iodine binding capacity is the Lambda650 (PE, Waltham, MA, USA), featuring a scanning range from 700 to 500 nm. Iodine binding capacity was computed by assessing the ratio of absorbance values at 635 nm and 520 nm, aligning with the wavelength of maximum absorbance. This calculation leveraged the peak absorption wavelength, ensuring accurate determination [63].

### 4.7. Statistical Analysis

Microsoft Excel 2016 (Microsoft, Redmond, WA, USA) and SPSS (version 26, IBM, Armonk, NY, USA) were employed for data processing, significance testing, and ANOVA, and when *p* < 0.05, multiple comparisons were performed using Tukey, and the marked letter method was used to indicate differences between treatments. Graphs were plotted with OriginPro (version 2022, OriginLab, Northampton, MA, USA).

## 5. Conclusions

The results of this study indicated that the conventional fertilization method with additional S application significantly increased the grain weight of fresh waxy maize, as well as its starch and protein content. Specifically, applying a basal dose of 225/75/75/75 kg ha^−1^ (N/P_2_O_5_/K_2_O/S) at sowing enhanced the pasting and thermal properties of waxy maize flour and starch. Therefore, we recommend the judicious addition of sulfur fertilizer in fresh waxy maize production, or incorporating sulfur into compound fertilizers, to achieve high-yield and high-quality cultivation of fresh waxy maize, which not only can increase farmers’ incomes, but also maintain the nutrient balance of the farmland, promoting the sustainable development of agriculture.

## Figures and Tables

**Figure 1 plants-13-02677-f001:**
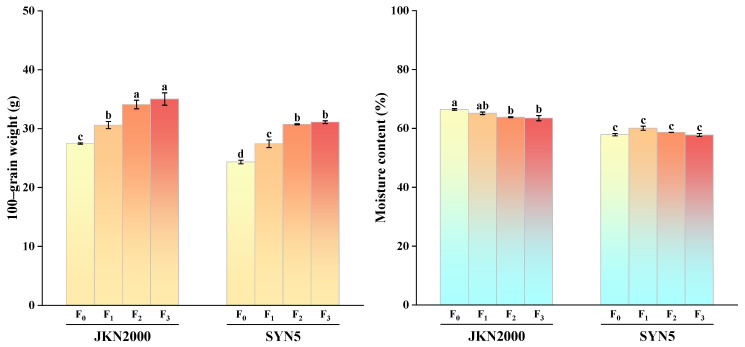
Effects of sulfur fertilizer application on 100-grain weight and moisture content of fresh waxy maize. JKN2000: Jingkenuo2000; SYN5: Suyunuo5; F_0_: no fertilizer; F_1_: application of 75 kg S ha^−1^; F_2_: applications of 225, 75, and 75 kg N, P, and K ha^−1^; F_3_: applications of 225, 75, 75, and 75 kg N, P, K and S ha^−1^. The different letters on the columns mean the difference was significant at the 0.05 probability level. The figure shows the mean values of three replicates of the same treatment.

**Figure 2 plants-13-02677-f002:**
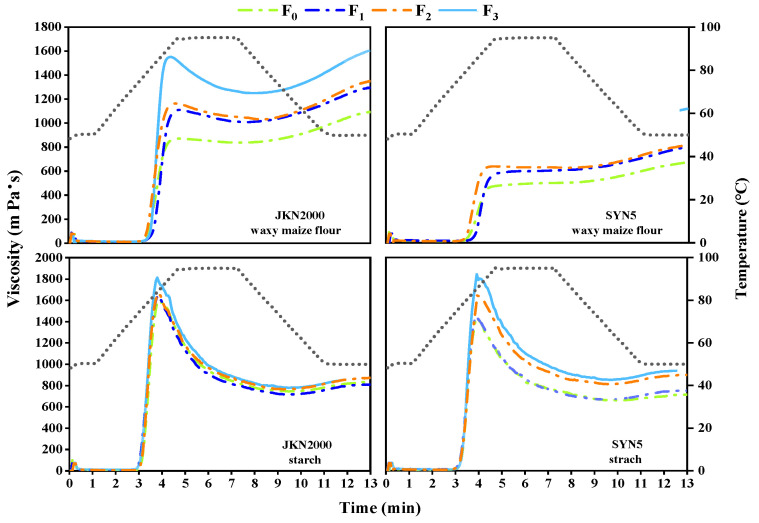
Effects of sulfur application on pasting properties of fresh waxy maize. JKN2000: Jingkenuo2000; SYN5: Suyunuo5; F_0_: no fertilizer; F_1_: application of 75 kg S ha^−1^; F_2_: applications of 225, 75, and 75 kg N, P, and K ha^−1^; F_3_: applications of 225, 75, 75, and 75 kg N, P, K, and S ha^−1^. The figure shows the mean values of three replicates with the same treatment. The gray dotted line in the legend indicates the temperature represented on the right y-axis.

**Figure 3 plants-13-02677-f003:**
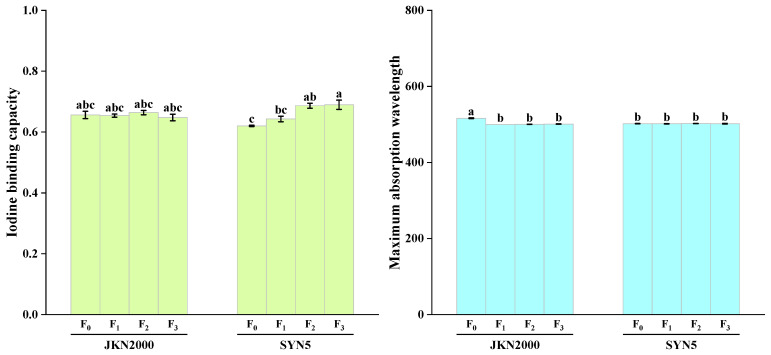
Effects of sulfur application on iodine binding capacity and maximum absorption wavelength of fresh waxy maize. JKN2000: Jingkenuo2000; SYN5: Suyunuo5; F_0_: no fertilizer; F_1_: application of 75 kg S ha^−1^; F_2_: applications of 225, 75, and 75 kg N, P, and K ha^−1^; F_3_: applications of 225, 75, 75, and 75 kg N, P, K, and S ha^−1^. The different letters on the column mean the difference was significant at the 0.05 probability level. The figure shows the mean values of three replicates with the same treatment.

**Figure 4 plants-13-02677-f004:**
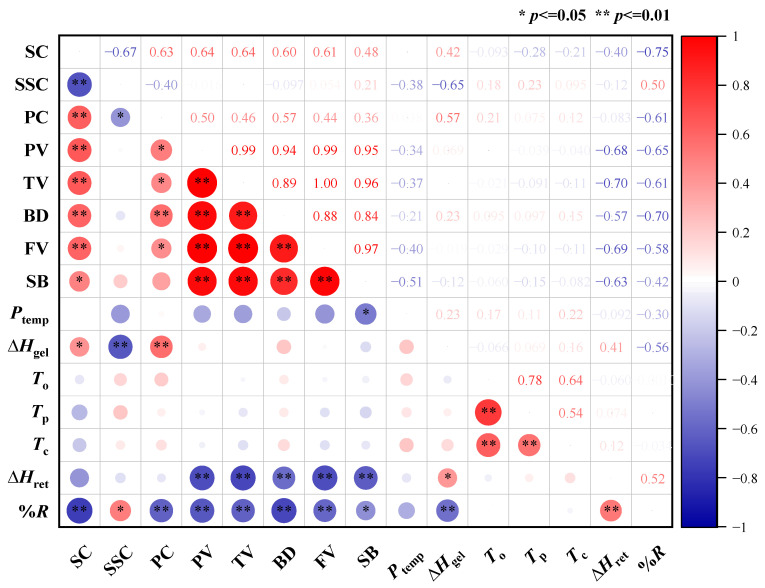
Correlation analysis of grain quality parameters. SC: starch content; SSC: soluble sugar content; PC: protein content; PV: peak viscosity; TV: trough viscosity; BD: breakdown viscosity; FV: final viscosity; SB: setback viscosity; *P*_temp_: pasting temperature; ∆*H*_gel_: gelatinization enthalpy; *T*_o_: onset temperature; *T*_p_: peak gelatinization temperature; *T*_c_: conclusion temperature; ∆*H*_ret_: retrogradation enthalpy; *%R*: retrogradation percentage. * and ** represent significance at the 0.05 and 0.01 probability levels.

**Figure 5 plants-13-02677-f005:**
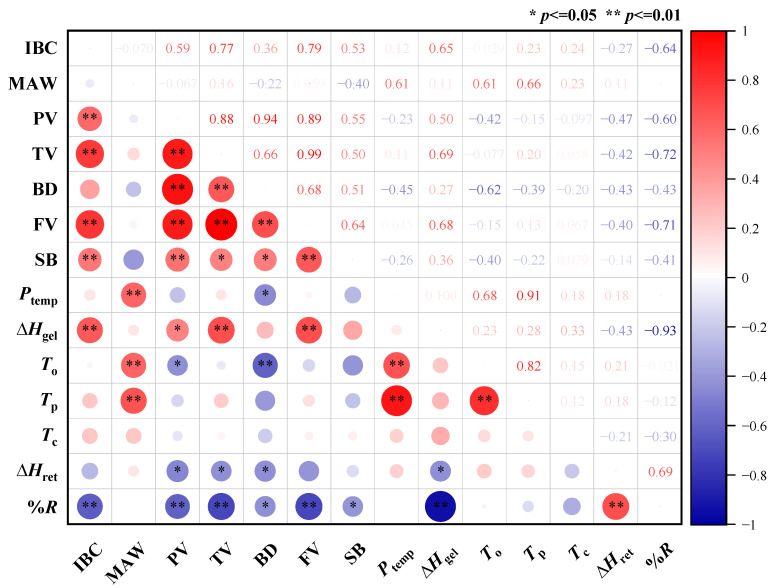
Correlation analysis of starch quality parameters. IBC: iodine binding capacity; MAW: maximum absorption wavelength; PV: peak viscosity; TV: trough viscosity; BD: breakdown viscosity; FV: final viscosity; SB: setback viscosity; *P*_temp_: pasting temperature; ∆*H*_gel_: gelatinization enthalpy; *T*_o_: onset temperature; *T*_p_: peak temperature; *T*_c_: conclusion temperature; ∆*H*_ret_: retrogradation enthalpy; *%R*: retrogradation percentage. * and ** represent significance at the 0.05 and 0.01 probability levels.

**Figure 6 plants-13-02677-f006:**
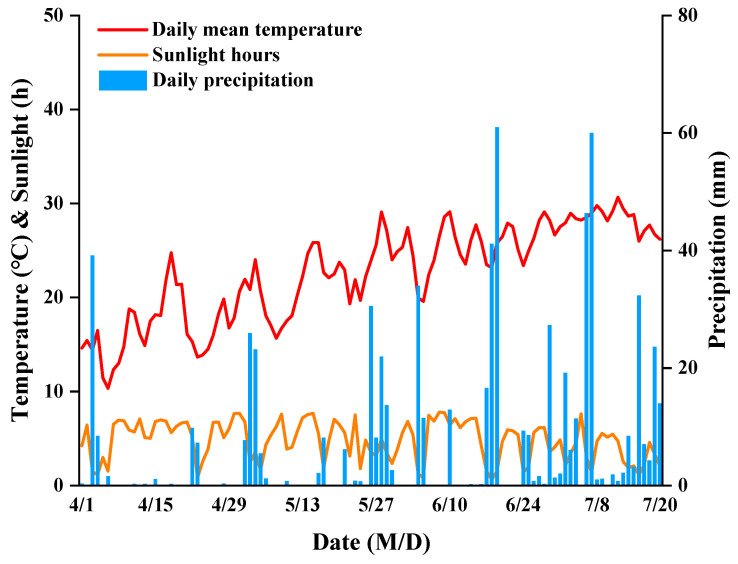
Daily mean temperature, sunlight hours, and precipitation during maize growth seasons in 2023.

**Table 1 plants-13-02677-t001:** Effects of sulfur application on grain component contents of fresh waxy maize.

Variety	Treatment	Starch (mg g^−1^)	Soluble Sugar (mg g^−1^)	Protein (mg g^−1^)
JKN2000	F_0_	559.6 c	87.2 a	89.8 c
	F_1_	571.4 b	86.5 a	91.9 bc
	F_2_	584.0 a	74.7 b	97.2 abc
	F_3_	586.7 a	71.0 c	103.1 a
SYN5	F_0_	551.8 d	84.2 a	92.9 bc
	F_1_	562.3 c	74.5 b	89.2 c
	F_2_	579.7 a	60.2 d	94.9 abc
	F_3_	586.2 a	59.7 d	99.8 ab
F-value				
	V	12.5 **	158.8 **	0.4
	T	84.9 **	150.8 **	5.7 **
	V×T	1.6	9.6 **	0.5

JKN2000: Jingkenuo2000; SYN5: Suyunuo5; F_0_: no fertilizer; F_1_: application of 75 kg S ha^−1^; F_2_: applications of 225, 75, and 75 kg N, P, and K ha^−1^; F_3_: applications of 225, 75, 75, and 75 kg N, P, K, and S ha^−1^. Values followed by different small letters within a column are significantly different at the 0.05 probability level. ** represent significance at the 0.05 and 0.01 probability levels.

**Table 2 plants-13-02677-t002:** Effects of sulfur application on pasting properties of fresh waxy maize.

Variety	Treatment	Waxy Maize Flour						Starch					
		PV (mPa·s)	TV (mPa·s)	BD (mPa·s)	FV (mPa·s)	SB (mPa·s)	*P*_temp_ (°C)	PV (mPa·s)	TV (mPa·s)	BD (mPa·s)	FV (mPa·s)	SB (mPa·s)	*P*_temp_ (°C)
JKN2000	F_0_	905.7 d	866.0 d	39.7 e	1137.3 d	271.3 b	78.3 c	1556.3 d	732.7 c	823.7 c	819.0 d	86.3 bc	75.1 c
	F_1_	1092.0 c	1000.7 c	91.3 c	1287.7 c	287.0 b	81.3 b	1623.7 c	712.0 d	911.7 b	805.0 d	93.0 bc	74.1 de
	F_2_	1194.0 b	1061.3 b	132.7 b	1382.3 b	321.0 a	78.1 c	1697.3 b	769.3 b	928.0 b	878.0 c	108.7 a	74.6 cd
	F_3_	1556.7 a	1249.3 a	307.3 a	1589.0 a	339.7 a	80.5 b	1781.0 a	777.7 b	1003.3 a	858.7c	81.0 c	74.0 e
SYN5	F_0_	501.0 f	494.0 f	7.0 e	671.7 f	177.7 e	81.2 b	1418.3 e	651.3 f	767.0 d	712.0 f	60.7 d	76.2 ab
	F_1_	582.7 e	576.3 e	6.3 e	781.0 e	204.7 d	82.9 a	1430.3 e	673.7 e	756.7 d	754.7 e	81.0 c	75.9 b
	F_2_	621.3 e	615.0 e	6.3 e	794.3 e	179.3 e	80.8	1670.3 bc	824.0 a	846.3 c	910.3 b	86.3 bc	75.9 b
	F_3_	935.0 d	860.3 d	74.7 c	1089.0 d	228.7 c	83.3 a	1794.3 a	843.3 a	951.0 b	939.7 a	96.3 ab	76.8 a
F-value													
	V	1176.0 **	1377.9 **	386.8 **	1143.8 **	446.7 **	78.4 **	44.3 **	0.1	65.4 **	5.5 *	12.7 **	151.3 **
	T	225.8 **	205.4 **	150.5 **	143.0 **	23.7 **	23.7 **	116.0 **	176.7 **	54.2 **	235.3 **	10.0 **	4.0 *
	V×T	9.3 **	2.3	50.0 **	2.9	6.5 **	1.2	13.7 **	54.6 **	4.9 *	80.1 **	8.8 **	7.3 **

JKN2000: Jingkenuo2000; SYN5: Suyunuo5; F_0_: no fertilizer; F_1_: application of 75 kg S ha^−1^; F_2_: applications of 225, 75, and 75 kg N, P, and K ha^−1^; F_3_: applications of 225, 75, 75, and 75 kg N, P, K, and S ha^−1^; PV: peak viscosity; TV: trough viscosity; BD: breakdown viscosity; FV: final viscosity; SB: setback viscosity; *P*_temp_: pasting temperature. Values followed by different small letters within a column are significantly different at the 0.05 probability level. * and ** represent significance at the 0.05 and 0.01 probability levels.

**Table 3 plants-13-02677-t003:** Effects of sulfur application on thermal properties of fresh waxy maize.

Variety	Treatment	Waxy Maize Flour						Starch					
		*∆H*_gel_ (J g^−1^)	*T*_o_ (°C)	*T*_p_ (°C)	*T*_c_ (°C)	*∆H*_ret_ (J g^−1^)	%*R* (%)	*∆H*_gel_ (J g^−1^)	*T*_o_ (°C)	*T*_p_ (°C)	*T*_c_ (°C)	*∆H*_ret_ (J g^−1^)	%*R* (%)
JKN2000	F_0_	10.4 cd	72.0 ab	79.3 ab	85.3 bc	5.3 abc	50.8 a	19.7 cd	65.0 d	70.8 e	83.2 abc	8.1 a	41.1 ab
	F_1_	8.8 d	75.8 a	79.9 a	86.1 abc	3.9 e	44.3 b	22.0 bc	63.9 e	69.8 f	87.9 ab	8.0 a	36.8 b
	F_2_	10.9 bc	71.7 b	76.3 c	85.9 abc	4.9 cd	45.3 ab	24.3 b	65.6 c	71.2 d	79.1 bc	8.0 a	33.7 bc
	F_3_	12.8 a	74.9 ab	80.7 a	87.4 a	4.5 de	34.7 c	22.0 bc	64.8 d	70.5 e	74.5 c	7.9 a	36.5 b
SYN5	F_0_	11.4 abc	75.1 ab	80.8 a	87.1 ab	5.8 a	50.6 a	17.8 d	67.4 b	73.0 c	80.8 abc	8.7 a	49.2 a
	F_1_	10.8 bc	72.9 ab	78.0 b	87.2 a	5.3 abc	49.0 ab	22.0 bc	67.0 b	73.3 bc	82.5 abc	8.4 a	38.3 b
	F_2_	12.3 ab	74.6 ab	79.9 a	86.3 abc	5.6 ab	45.4 ab	29.8 a	67.8 a	73.5 b	89.3 a	7.9 a	26.7 c
	F_3_	13.0 a	72.1 ab	77.9 b	85.2 c	5.0 bcd	38.0 c	24.2 b	65.7 c	74.0 a	81.8 abc	8.0 a	33.3 bc
F-value													
	V	9.5 **	0.1	0.1	0.6	27.6 **	2.2	2.5	452.8 **	3500.7 **	1.7	1.5	0.1
	T	12.3 **	0.4	5.0	0.4	10.0 **	20.6 **	14.4 **	45.3 **	63.3 **	2.8	0.7	9.2 **
	V×T	0.9	4.0	17.8 **	5.3 *	1.9	0.8	3.1	20.5 **	52.6 **	4.1 *	0.4	2.5

JKN2000: Jingkenuo2000; SYN5: Suyunuo5; F_0_: no fertilizer; F_1_: application of 75 kg S ha^−1^; F_2_: applications of 225, 75, and 75 kg N, P, and K ha^−1^; F_3_: applications of 225, 75, 75, and 75 kg N, P, K, and S ha^−1^; ∆*H*_gel_: gelatinization enthalpy; *T*_o_: onset temperature; *T*_p_: peak gelatinization temperature; *T*_c_: conclusion temperature; ∆*H*_ret_: retrogradation enthalpy; %*R*: retrogradation percentage. Values followed by different small letters within a column are significantly different at the 0.05 probability level. * and ** represent significance at the 0.05 and 0.01 probability levels.

## Data Availability

Informed consent was obtained from all subjects involved in the study. All the data and code used in this study can be requested by email to the corresponding author, Guanghao Li, at guanghaoli@yzu.edu.cn.

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
