# Peer review of "Effects of Sulfur Application on the Quality of Fresh Waxy Maize"

_plants, 2024, doi:10.3390/plants13192677_

Round 1

Reviewer 1 Report

Comments and Suggestions for Authors

Please find the comments attached

Author Response

Reviewer #1

Comments to authors:

In this manuscript, the author studied the effect of sulphur on the quality of waxy maize. The study objective is interesting and the findings reported in this manuscript will advance the existing knowledge on the role of sulphur in plants. However, I have some observations as indicated below.

Abstract

  1. Line 13-14: The description of the methods is adequate for an abstract but could be improved by specifying the experimental conditions, such as the location or time frame.

Reply: Thanks for your valuable comments. The original sentence has been revised to: Field experiments were conducted in Jiangsu, Yangzhou, China, from April 1 to July 20 in 2023. Jingkenuo2000 (JKN2000) and Suyunuo5 (SYN5) were used as experiment materials, and four treatments were set: no fertilizer application (F0), S fertilizer application (F1), conventional fertilization method (F2), and conventional fertilization method with additional S application (F3).

  1. Line 19: The results could be presented more clearly, particularly the percentages and their comparison. For instance, instead of "the increase was characterized by F3>F2>F1," it could specify "F3resulted in the highest increase, followed by F2 and F1."

Reply: Thanks for your valuable comments. The original sentence has been revised to: Among these, F3 exhibited the most significant increases. Specifically, in JKN2000, the grain weight, starch content, and protein content increased by 27.7%, 4.8%, and 14.8%, respectively, while in SYN5, these parameters increased by 26.3%, 6.2%, and 7.4%, respectively, followed by F2 and F1

  1. Line 31: Keywords should be arranged alphabetically

Reply: Thanks for your valuable comments. Keywords have been arranged alphabetically in the revised manuscript.

Introduction

  1. The concept of sulfur deficiency and its impact on crops, especially on maize, is repeated several times. So, it is better to avoid redundancy. You can combine these points to succinctly explain that sulfur deficiency is a growing concern due to environmental changes and has varying impacts on different crops, including maize.

Reply: Thanks for your valuable comments. The necessary revisions have been made in the manuscript. The original sentence has been revised to: Due to reduced atmospheric sulfur inputs, stricter environmental regulations, and changes in fertilization practices, sulfur deficiency is becoming a growing concern, and has varying impacts on different crops, including maize.

  1. Line 34-35: could be rephrased to "Waxy maize, also known as sticky maize, is valued for its rich nutritional content and palatability, and it covers over 800,000 hectares in China."

Reply: Thanks for your valuable comments. According to your suggestion, this sentence has been revised. Waxy maize, also known as sticky maize, is valued for its rich nutritional content and palatability, and it covers over 800,000 hectares in China.

  1. Line 38: The phrase "with living standards improve" should be corrected to "as living standards improve." Additionally, "were expanding" should be in the present tense ("are expanding") to maintain consistency.

Reply: Thanks for your valuable comments. The phrase "with living standards improve" has been revised to "as living standards improve.", "were expanding" has been revised to "are expanding".

  1. Line 42-43: This is a strong transition that highlights the problem. Please consider using the present tense ("is becoming") for consistency. Additionally, specify how these changes in fertilization practices contribute to sulfur deficiency if space allows.

Reply: Thanks for your valuable comments. The phrase "was becoming" has been revised to "is becoming ". In modern agriculture, N fertilizers are often applied in large amounts to boost crop yields. However, an imbalance where N application greatly exceeds sulfur can lead to relative sulfur deficiency. While excessive N increases crop nitrogen uptake, it does not proportionally enhance sulfur availability, resulting in a relative sulfur deficit. Additionally, soil type and environmental conditions play a significant role in sulfur supply. Soils with inherently low sulfur levels can lead to deficiencies if fertilization practices are not adjusted accordingly. Environmental factors, including precipitation and soil pH, also impact sulfur availability and utilization.

  1. Line 61-62: While the fertilization details might be useful, they could be overwhelming in the introduction. You can summarize the key findings without listing specific quantities.

Reply: Thanks for your valuable comments. The necessary revisions have been made in the revised manuscript. Excessive details about fertilization have been eliminated.

  1. Line 67-68: “S application can facilitate the movement of photosynthetic products from leaves to bulbs in garlic, thereby boosting its yield without compromising its nutritional quality [17]." This example about garlic is interesting but not directly relevant to maize. Consider omitting or replacing it with a maize-specific example.

Reply: Thanks for your valuable comments. The example about garlic has been omitted.

  1. Line 69-71: Mentions of garlic and wheat could be seen as unnecessary unless they are directly tied to waxy maize. If these examples are not critical to understanding the effects on waxy maize, please omit or replace them with examples more directly related to maize.

Reply: Thanks for your valuable comments. The example about wheat has been omitted. The original sentence has been revised to: The combined application of N and S fertilizers enhanced nitrogen uptake in maize, increasing yield, protein content, and starch content in the grains[17].

  1. Line 78: There's a significant focus on nitrogen fertilization, which, while relevant, could overshadow the primary focus on sulfur. Balance the discussion by slightly reducing the emphasis on nitrogen, especially where it repeats information already covered, and instead highlight the specific interactions between nitrogen and sulfur in waxy maize.

Reply: Thanks for your valuable comments. The necessary revisions have been made in the revised manuscript. Although N fertilizers are essential for increasing crop yield, optimal N levels im-prove the starch's gelatinization and thermodynamic properties. Conversely, excessive N application can significantly reduce PV and BD of waxy maize starch and increase pasting temperature.

  1. Line 87-89: These sentences are relevant but could be more directly tied to maize by discussing similarities or differences in how maize and buckwheat respond to N fertilization.

Reply: Thanks for your valuable comments. The original sentence has been revised to: Similarly, buckwheat exhibits a notable response to nitrogen fertilization. Research has demonstrated that optimal nitrogen application not only enhances buckwheat grain weight and PC but also improves starch quality[27,28].

  1. Line 97-98: The statement about N being emphasized over S is crucial but could be expanded with a brief explanation of why this imbalance occurs.

Reply: Thanks for your valuable comments. A brief explanation has been added to highlight the increased emphasis on N fertilizers relative to S fertilizers: N fertilizers are commonly prioritized over sulfur in glutinous maize production due to their proven ability to boost yields and deliver significant short-term economic benefits.

  1. Line 98: Consider rephrasing "was often arbitrary" to "often lacked precision," to convey the issue more accurately

Reply: Thanks for your valuable comments. The phrase "was often arbitrary" has been revised to "often lacked precision" in the revised manuscript.

  1. Line 108-109: The third objective could be made more specific. Instead of "to provide a reference for rational fertilization," it could be rephrased as "to develop guidelines for optimizing fertilization practices and improving the quality of fresh waxy maize in southern China.

Reply: Thanks for your valuable comments. The original sentence has been revised according to your suggestion.

Results

  1. Line 114-115: The term “minimal overall impact” could be clarified—if the moisture contentwas statistically insignificant or only slightly altered.

Reply: Thanks for your valuable comments. The original sentence has been revised to: Fertilization treatments resulted in significant increases in 100−grain weight of both varieties compared to F0, while the moisture content did not exhibit statistically significant differences.

  1. Line 115-118: improve clarity to state: "The 100-grain weight of JKN2000 increased by 11.3% under F1, 24.0% under F2, and 27.4% under F3. For SYN5, the increases were 12.7% under F1, 26.3% under F2, and 27.7% under F3. The largest increase was observed with the F3 treatment."

Reply: Thanks for your valuable comments. The original sentence has been revised according to your suggestion.

  1. Line 129-130: This provides a comparison between varieties. It is clear and precise but consider mentioning specific values or percentages for clarification.

Reply: Thanks for your valuable comments. The original sentence has been revised to: The SC, SSC, and PC of JKN2000 were 14.6%, 1.0%, and 1.4% higher, respectively, than those of SYN5, but the differences in protein was not statistically significant.

  1. Line 131-132: The contents of starch, soluble sugar, and protein in JKN2000 were higher than SYN5. It could be better to provide the percentage changes or specific values for clarity.

Reply: Thanks for your valuable comments. The original sentence has been revised to: The SC, SSC, and PC of JKN2000 were 14.6%, 1.0%, and 1.4% higher, respectively, than those of SYN5, but the differences in protein was not statistically significant.

  1. Line 144-145: It may be helpful to specify the magnitude of the difference between PV and BD, if known.

Reply: Thanks for your valuable comments. The original sentence has been revised to: The PV and BD of the starch were 75.6% and 950.3% higher, respectively, compared to those of waxy maize flour.

  1. Line 191-193: The sentence “The MAW of JKN2000 showed a consistent trend of significant decrease in F1, F2, and F3, and there was no significant difference among F1, F2, and F3.” is a bit complex. It might benefit from rephrasing to avoid redundancy. For example: "The MAW of JKN2000 significantly decreased across F1, F2, and F3 treatments, with no significant differences observed among these treatments."

Reply: Thanks for your valuable comments. The original sentence has been revised according to your suggestion.

Discussion

  1. The discussion can be better organized by grouping similar topics together. For example, discuss all findings related to starch properties in one section and those related to protein in another.

Reply: Thanks for your valuable comments. The discussion has been better organized by grouping similar topics together. The discussion was structured around three primary themes: component content in grains, pasting and thermal properties of waxy maize flour, pasting and thermal properties of starch.

  1. Line 231: The interaction between nitrogen and sulfur is discussed extensively, which is important, but it can overshadow other findings.

Reply: Thanks for your valuable comments. The necessary revisions have been made in the revised manuscript.

  1. Line 233-234: Grain weight was markedly increased which is a crucial finding. It might be beneficial to specify how much the grain weight increased in percentage terms to provide more context.

Reply: Thanks for your valuable comments. The original sentence has been revised to: Compared to no fertilizer application, all fertilization treatments markedly increased the grain weight of fresh waxy maize (F1 increased by 0.4%, F2 increased by 37.7%, and F3 increased by 54.2%).

  1. Line 234-235: The benefits of the conventional fertilization method with additional sulfur (F3) are repeated multiple times. This can be consolidated to avoid redundancy. Suggestion: Combine these discussions into a more concise statement, highlighting the superiority of the F3 treatment in improving multiple parameters.

Reply: Thanks for your valuable comments. These discussions have been combined into a more concise statement: Compared to no fertilizer application, all fertilization treatments markedly increased the grain weight of fresh waxy maize (F1 increased by 0.4%, F2 increased by 37.7%, and F3 increased by 54.2%), as well as the PC and SC in the grains. Notably, the conventional fertilization method with additional S application (F3) outperformed both the S fertilizer application (F1) and the conventional fertilization method (F2) in increasing these parameters.

  1. Line 237-238: This is a well-placed reference that strengthens your argument. Consider adding more detail about how their findings align with yours.

Reply: Thanks for your valuable comments. The necessary revisions have been added in the revised manuscript.

  1. Line 239-243: This section provides a solid mechanistic explanation. It might benefit from being broken into two sentences for readability. Also, linking these processes directly to increased grain weight could strengthen the argument.

Reply: Thanks for your valuable comments. The original sentence has been revised according to your suggestion. The application of S fertilizer increased glutathione levels in maize leaves, which reduced hydrogen peroxide accumulation[39]. This reduction helped maintain redox balance during photosynthesis[40], leading to an improved photosynthetic rate[41] and enhanced production of photosynthetic assimilates. The increase in photosynthetic products enhanced dry matter accumulation, which subsequently resulted in a substantial increase in maize grain weight.

  1. Line 245-247: This sentence “Maize grains were predominantly made up of starch, proteins, and soluble sugars, as 245 their proportions and intrinsic structures impacted the physicochemical properties of 246 waxy maize flour” is a good introduction to the composition of maize grains. However, it could be more concise. Consider rephrasing to: "Maize grains are primarily composed of starch, proteins, and soluble sugars, which influence the physicochemical properties of waxy maize flour."

Reply: Thanks for your valuable comments. The original sentence has been revised according to your suggestion.

  1. Line 251-253: You might consider citing specific studies that discuss cysteine crosslinking or disulfide bond formation to strengthen this point.

Reply: Thanks for your valuable comments. The specific study has been cited: The increase was likely due to S fertilizers promoting the cross–linking of cysteine and other amino acids, forming disulfide bonds that maintain protein stability and promote protein accumulation [45].

Laps, S.; Sun, H.; Kamnesky, G.; Brik, A. Palladium‐Mediated Direct Disulfide Bond Formation in Proteins Containing S‐Acetamidomethyl‐cysteine under Aqueous Conditions. Angewandte Chemie 2019, 131, 5785–5789.

  1. Line 257: You could also explicitly mention why F3was more effective

Reply: Thanks for your valuable comments. The necessary revisions have been made in the manuscript. Both N and S play essential roles in protein synthesis, with N serving as a primary building block of amino acids and S supporting the synthesis of key sulfur-containing amino acids like cysteine and methionine [48]. The synergistic application of these two elements not only optimizes the amino acid composition but also enhances protein quality. Furthermore, S facilitates N uptake and utilization in crops, improving N use efficiency and subsequently increasing protein content [49], so the conventional fertilization method with additional S application (F3) led to a more substantial increase in SC and PC than the S fertilizer application (F1) and the conventional fertilization method (F2).

  1. Line 264-65: The effect of S on wheat is a good point, but since your study is on maize, it might be useful to connect this more directly to maize by saying: "Similar effects might be expected in maize."

Reply: Thanks for your valuable comments. The sentence "similar effects might be expected in maize" has been added in the revised manuscript.

  1. Line 269-270: This is a clear and logical explanation. No major changes needed, but you could consider adding more context about why increased pasting viscosity is desirable.

Reply: Thanks for your valuable comments. The necessary revisions have been made in the revised manuscript. In food processing, higher viscosity helps in forming a uniform food structure. This leads to improved appearance and stability of the final product, as well as reduced layering or separation during storage and transportation.

  1. Line 288: The specific fertilization rates for rice starch are mentioned, but this might be too detailed and somewhat irrelevant to waxy maize. Consider removing specific fertilizer rates unless they directly relate to the study’s objectives or findings.

Reply: Thanks for your valuable comments. The specific fertilization rates for rice starch have been removed.

  1. Line 295-296: This is a strong explanation of the mechanism behind increased IBC. No major changes needed, but you could add a reference to support this mechanistic explanation if possible.

Reply: Thanks for your valuable comments. The reference has been added to support this mechanistic explanation: Yang, H.; Lu, D.; Shen, X.; Cai, X.; Lu, W. Heat Stress at Different Grain Filling Stages Affects Fresh Waxy Maize Grain Yield and Quality. Cereal Chemistry 2015, 92, 258–264.

Materials and methods

  1. Line 309: The experimental site could be improved by providing more specifics, such as the specific research station or farm if applicable, or geographical location

Reply: Thanks for your valuable comments. The original sentence has been revised to: Field experiments were conducted in 2023 at the Yangzhou University experimental farm in Yangzhou, Jiangsu Province, China (32.40° N, 119.43° E).

Conclusion

  1. Line 384-385: “The results of this study indicated that conventional fertilization method with additional S application significantly increased the grain weight and component content of fresh waxy maize”. Please specifying which components?

Reply: Thanks for your valuable comments. The original sentence has been revised to: The results of this study indicated that conventional fertilization method with additional S application significantly increased the grain weight of fresh waxy maize, as well as its starch and protein content.

  1. Line 388-390: Please mention the broader implications or benefits of this recommendation, such as its impact on farmer profitability, sustainability, or food quality.

Reply: Thanks for your valuable comments. The broader impacts and potential benefits of this recommendation are outlined below:

Therefore, we recommend the judicious addition of sulfur fertilizer in fresh waxy maize production, or incorporating sulfur into compound fertilizers, to achieve high–yield and high–quality cultivation of fresh waxy maize, which not only can increase farmers' income, but also maintain the nutrient balance of the farmland, promoting the sustainable development of agriculture.

Reviewer 2 Report

Comments and Suggestions for Authors

Why did the field experiments were conducted only 1 year? Due to weather conditions,  should be running  at least 2-3 years.

Author Response

Reviewer #2

Comments and Suggestions for Authors:

Why did the field experiments were conducted only 1 year? Due to weather conditions, should be running at least 2-3 years.

Reply: Thanks for your valuable comments. This experiment examined the effects of sulfur fertilizer on the grain weight and quality of fresh waxy maize, with a specific focus on grain and starch quality. The experiment was conducted in 2023 under optimal growing conditions, with no significant stress factors such as waterlogging, drought, heat injury, pests, or diseases affecting the growth period. Each treatment included three random replicates, and consistent trends across treatments demonstrated strong reproducibility. The results clearly demonstrated that adding sulfur fertilizer to standard fertilization practices increased the grain weight of waxy maize, and enhanced its iodine binding capacity (reflecting the chain length distribution of starch), and significantly improved the pasting and thermal properties of waxy maize flour and starch. Although it was only a one-year experiment, the results were very clear. Thanks for your valuable comments, we will continue to conduct relevant research in this direction from the perspectives of molecular and physiological mechanisms.

Additionally, some papers published in this journal revealed many similar field studies conducted over a single growing season, such as:

  1. Hua, L.; Yang, Z.; Li, W.; Zhao, Y.; Xia, J.; Dong, W.; Chen, B. Effects of different straw return modes on soil carbon, nitrogen, and greenhouse gas emissions in the semiarid maize field. Plants.2024, 13, 2503.
  2. Batyrbek, M.; Abbas, F.; Fan, R.; Han, Q. Influence of mineral fertilizer and manure application on the yield and quality of maize in relation to intercropping in the southeast republic of kazakhstan. Plants (Basel).2022, 11, 2644.

Reviewer 3 Report

Comments and Suggestions for Authors

Dear Chief-in-Editor,

Thank you for allowing me to review the manuscript entitled “Effects of sulfur application on the Quality of fresh waxy maize “.

The presented work deals with the effects of sulfur (S) application on the quality and characteristics of fresh waxy maize. The study examines the impact of different fertilization treatments, including sulfur application, on grain weight, starch, and protein content, iodine binding capacity, pasting properties, and thermal properties of waxy maize varieties (Jingkenuo2000 and Suyunuo5). and how adding sulfur to conventional fertilization methods influences the quality and nutritional content of fresh waxy maize.

The manuscript is generally well-written; however, in the following, I provide some in-depth comments, criticisms, worries, and recommendations that should be taken into account before a final judgment on the document is made.

Sincerely

The main criticism points are:                     

-           There are many grammatical and formatting errors in the whole article; check and modify it, please

-           In materials and methods: The authors briefly mentioned meteorological conditions (like total rainfall, temperature, and sunlight hours), but did not mention other environmental factors that could influence maize growth, such as soil pH, humidity levels, and possible pest or disease pressure.

-           Authors didn't describe the method by which they applied sulfur (e.g., granules, liquid)

-           Is there any additional fertilization or irrigation that occurred during the growth period??

-           The experiment included three replicates per treatment, there is no mention of whether these replicates were randomized or how they were distributed across the field. Explain in detail, please

-           The methods didn't mention any controls for other variables that could affect the results, such as water management, pest control, or other agronomic practices.

-           Describe how samples were handled, stored, or transported before analysis????? This is important because improper storage or handling (e.g., exposure to moisture or heat) could alter the composition of maize samples, potentially affecting the outcomes of the study.

-           In discussion: Correlation analysis revealed that the starch content (SC) positively correlated with protein content (PC), peak viscosity (PV), breakdown (BD), final viscosity (FV), setback (SB), and enthalpy of gelatinization (ΔHgel), but negatively correlated with soluble sugar content (SSC) and retrogradation percentage (%R). Also, soluble sugar content (SSC) positively correlated with %R and negatively correlated with PC and ΔHgel…  Peak viscosity (PV), breakdown (BD), final viscosity (FV), and protein content (PC) were positively correlated with each other.

This analysis suggests that sulfur fertilizer application not only improves grain weight and quality but also significantly alters the starch and therm. Please, compare your findings with the results of the previously published literature. Also, discuss the possible mechanisms.

-           Figures and tables are enough.

Comments on the Quality of English Language

Minor editing of English language required.

Author Response

Reviewer #3

  1. There are many grammatical and formatting errors in the whole article; check and modify it, please.

Reply: Thanks for your valuable comments. We invited an English native speaker to check the language of the paper. The grammatical and formatting errors in the article have been reviewed and corrected.

  1. In materials and methods: The authors briefly mentioned meteorological conditions (like total rainfall, temperature, and sunlight hours), but did not mention other environmental factors that could influence maize growth, such as soil pH, humidity levels, and possible pest or disease pressure.

Reply: Thanks for your valuable comments. The necessary revisions have been made in the revised manuscript. The test soil under examination predominantly exhibited a silt loam texture and the PH was 6.2. At sowing, all fertilizers were uniformly applied as basal dressing, and other agronomic practices (water management, pest and disease control) followed high–yield management protocols. The experiment was conducted in 2023 under optimal growing conditions, with no significant stress factors such as waterlogging, drought, heat injury, pests, or diseases affecting the growth period.

  1. Authors didn't describe the method by which they applied sulfur (e.g., granules, liquid)

Reply: Thanks for your valuable comments. The field experiment employed compound fertilizer (N−P2O5−K2O=27−9−9, provided by Jiangsu Zhongdong Fertilizer Co., Ltd.) and sulfur fertilizer (sulfur content 95%, bought from Tiger-Sul, Inc.) as sources of N, P, K, and S fertilizers. The sulfur fertilizer was in the form of solid spherical granules (see the figure below) and was applied once as a basal dose at sowing.

  1. Is there any additional fertilization or irrigation that occurred during the growth period?

Reply: Thanks for your valuable comments. No additional fertilization or irrigation was provided during the maize growth period. Fertilizers were applied once as a basal dose at sowing, and water supply depended entirely on rainfed conditions throughout the growth cycle, and no additional irrigation was provided during the maize growth period in 2023.

  1. The experiment included three replicates per treatment, there is no mention of whether these replicates were randomized or how they were distributed across the field. Explain in detail, please

Reply: Thanks for your valuable comments. The three replicates for each treatment were randomized, and their field distribution was shown in the diagram below. Pollination bags were used during the fresh-eating stage of waxy maize to prevent cross-pollination between different varieties.

  1. The methods didn't mention any controls for other variables that could affect the results, such as water management, pest control, or other agronomic practices.

Reply: Thanks for your valuable comments. The necessary revisions have been made in the revised manuscript. At sowing, all fertilizers were uniformly applied as basal dressing, and other agronomic practices (water management, pest and disease control) followed high–yield management protocols. The experiment was conducted in 2023 under optimal growing conditions, with no significant stress factors such as waterlogging, drought, heat injury, pests, or diseases affecting the growth period.

  1. Describe how samples were handled, stored, or transported before analysis????? This is important because improper storage or handling (e.g., exposure to moisture or heat) could alter the composition of maize samples, potentially affecting the outcomes of the study.

Reply: Thanks for your valuable comments. Each treatment selected three uniformly developed maize ears, from which grains were carefully extracted from the middle and thoroughly mixed. Subsequently, 200 g of grains were randomly sampled for the preparation of starch and waxy maize flour samples. The grains (100 g) were steeped in 500 mL of 1g L−1 NaHSO3 solution for 48h at room temperature. Following this, starch separation was carried out following established protocols from previous studies[1]. The grains (100 g) were subjected to drying in an oven (TENGREN DZ47−63) set at 60°C until reaching a constant weight. After drying, the grains were finely ground using a grinder (RS−FS1406) and passed through a 100−mesh sieve. The resulting grain flour was utilized for analyzing grain flour component content, gelatinization, and thermodynamic properties.

The waxy maize flour and starch samples were first placed into 50 mL centrifuge tubes, sealed in airtight bags, and stored in a cool, dry environment. To minimize the effects of moisture and temperature, the samples were quickly transported to the laboratory for timely measurement and analysis.

[1] Lu, D.; Lu, W. Effects of protein removal on the physicochemical properties of waxy maize flours. Starch - Stärke. 2012, 64, 874–881.

  1. In discussion: Correlation analysis revealed that the starch content (SC) positively correlated with protein content (PC), peak viscosity (PV), breakdown (BD), final viscosity (FV), setback (SB), and enthalpy of gelatinization (ΔHgel), but negatively correlated with soluble sugar content (SSC) and retrogradation percentage (%R). Also, soluble sugar content (SSC) positively correlated with %R and negatively correlated with PC and ΔHgel…  Peak viscosity (PV), breakdown (BD), final viscosity (FV), and protein content (PC) were positively correlated with each other.This analysis suggests that sulfur fertilizer application not only improves grain weight and quality but also significantly alters the starch and therm. Please, compare your findings with the results of the previously published literature. Also, discuss the possible mechanisms.

Reply: Thanks for your valuable comments. The necessary revisions have been made in the revised manuscript. Correlation analysis of waxy maize flour quality showed that SC was significantly positively correlated with ΔHgel, and significantly negatively correlated with %R. This indicated that high SC was conducive to a high ΔHgel (better thermal stability). Under the experimental conditions, fertilization treatments led to a significant increase in starch content in both maize varieties. Waxy maize starch, consisting of nearly 100% amylopectin, requires more energy to disrupt and gelatinize its highly ordered structure. This results in greater thermal stability and improved anti-retrogradation properties. Consequently, sulfur application in this study enhanced the starch content of waxy maize, leading to an increase in ΔHgel. Previous research showed that waxy maize flour with high ΔHgel often formed higher pasting viscosity after gelatinization. This study had similar results, indicating that a high ΔHgel was associated with stronger hydrogen bonds in starch granules, necessitating more energy to break these bonds during gelatinization and resulting in higher pasting viscosity.

Correlation analysis of starch quality showed that IBC was significantly positively correlated with pasting viscosities (PV, TV, FV) and ΔHgel., and significantly negatively correlated with %R. This suggests that a higher IBC enhances starch's swelling and shear resistance, leading to increased pasting viscosity and improved an-ti-retrogradation properties. Under the conditions of this experiment, SYN5 exhibited a more significant response to fertilization regarding IBC, with the highest value recorded under the F3 treatment. The increased IBC may result from fertilization, which reduced the activity of starch branching enzymes. This reduction led to decreased branching frequency in amylopectin, resulting in a higher proportion of long chains. These longer chains could form more stable complexes with iodine, thereby enhancing the IBC, amylopectin with long chains has higher pasting viscosity after gelatinization, higher ΔHgel, better thermal stability, and stronger anti–retrogradation ability.

Round 2

Reviewer 1 Report

Comments and Suggestions for Authors

The authors have reivised the manuscript as per my suggestions. I am happy with the revision.

Reviewer 3 Report

Comments and Suggestions for Authors

Authors responded well to raised comments

Comments on the Quality of English Language

No comments